# Biopriming of *Cucumis sativus* L. Seeds with a Consortium of Nitrofixing Cyanobacteria Treated with Static Magnetic Field

**DOI:** 10.3390/plants14040628

**Published:** 2025-02-19

**Authors:** Yadenis Ortega Díaz, Liliana Gómez Luna, Yilan Fung Boix, Yadira Silveira Font, Els Prinsen, Michiel Huybrechts, Dries Vandamme, Ann Cuypers

**Affiliations:** 1National Center for Applied Electromagnetism, Santiago de Cuba 90600, Cuba; yortega@uo.edu.cu (Y.O.D.); lilianag@uo.edu.cu (L.G.L.); fungboix26@gmail.com (Y.F.B.); yadira89066@gmail.com (Y.S.F.); 2Department of Biology, University of Antwerp, Groenenborgerlaan 171, 2020 Antwerp, Belgium; els.prinsen@uantwerpen.be; 3Environmental Biology, Centre for Environmental Sciences, Hasselt University, 3590 Diepenbeek, Belgium; michiel.huybrechts@uhasselt.be; 4Analytical and Circular Chemistry, Center for Enveriomental Sciences (CMK), Institute for Materials Research (IMO), Hasselt University, 3590 Diepenbeek, Belgium

**Keywords:** biopriming, consortium nitrofixing cyanobacteria, *Cucumis sativus* L., static magnetic field

## Abstract

The growing demand for sustainable agriculture necessitates innovative strategies to enhance crop productivity while minimizing environmental impact. This study explores the biopriming potential of *Cucumis sativus* L. seeds using extracts derived from a consortium of nitrofixing cyanobacteria *Nostoc commune*, *Calothrix* sp., and *Aphanothece minutissima* subjected to static magnetic field (SMF) treatments. The cyanobacterial consortia were exposed to SMF at varying magnetic inductions (40–50 mT and 100–200 mT), followed by extract preparation and application as biopriming agents. Results demonstrated significant improvements in key seedling growth parameters, including root and stem length, vigor index I, and fresh biomass. The consortium treated with 40–50 mT SMF showed the most pronounced growth-stimulating activity, suggesting enhanced bioactive compound production under this treatment that might be related to auxin biosynthesis. Biopriming with cyanobacterial extracts maintained a balanced nutritional uptake and plant health, as indicated by stable fresh weight dry weight ratios. These findings highlight the potential of SMF-enhanced cyanobacterial consortia as biopriming agents for horticultural crops. Future research should elucidate the underlying modes of action and optimize conditions for broader crop applications.

## 1. Introduction

Intensive agriculture using chemical fertilizers has increased crop productivity in recent decades, but at the same time, this has caused global detrimental impacts on the environment, such as increased soil salinity, accumulation of heavy metals, eutrophication of water, and nitrate accumulation [1,2]. This situation is aggravated by rapid population growth, which will require a 60% increase in agricultural production in the next 30 years [3,4]. In addition, a changing climate, including higher temperatures, droughts, more extreme events, and sea-level rise, is making the task of increased food production much more difficult as it negatively affects the quantity as well as the quality of our food supplies [5]. Climate change has already affected global food production, leading to a decline in the productivity and projected yields of major staple food crops across all regions of the world. This poses a growing threat to both current and future agricultural production [5]. As a consequence, maintaining high agricultural productivity, mitigating environmental impact, and promoting environmental regeneration is an urgent challenge. In this sense, seed germination is a key developmental transition in a plant’s life, essential for its successful establishment. Therefore, it needs to be considered in future crop production programs to ensure optimal plant growth and crop yield [6,7,8].

The ability of seeds to germinate is a key determinant in ensuring crop production. Seed germination is the onset of the plant’s life cycle, and it not only determines the number of potentially viable plants but also their ability to withstand environmental stresses, such as climate change. In this context, seed vigor is a complex agronomic trait, which includes seed longevity, germination rate, and seedling growth, as well as early stress resilience. Hence, it determines the success of initial establishment in plant development [9]. Seed aging has been well recognized as the major cause of reduced vigor and a decline in germination [10]. Moreover, seeds generally lose their germination value due to various factors, such as climate change, which results in the cultivation being limited to a short period of time, thus affecting the morphology and growth of the plants and ultimately limiting their productivity [9]. During germination, many biochemical processes occur due to a constantly operating network of genes and hormones [11]. The speed, uniformity, and quality of seed germination significantly influence the subsequent growth and condition of the plant. Therefore, optimizing seed germination strategies is essential in maximizing agricultural productivity and ensuring food security [9]. In this regard, physical pretreatment, priming, and, more specifically, biopriming are techniques used to improve seed germination. Research on these topics and on the underlying working mechanisms have gained importance over the last decades.

There are several physical seed pretreatment methods that stimulate germination and seedling growth. As such, plasma treatment on barley (*Hordeum vulgare* L.) seeds enhanced seed germination and seedling growth across different stress conditions, such as drought, salinity, and cold. In addition, the obtained results demonstrated that plasma treatment affected the plant’s biochemistry, supporting plant resilience under stress. Changes in the concentration of biochemical growing factors led to faster germination and initially more robust plant growth, even under stress conditions [12]. Other studies highlighted the beneficial effects of other physical treatments on the seed germination rate and vigor. Positive biometrical effects of pre-sowing laser stimulation on the germination and early growth of Sainfoin seeds (*Onobrychis viciifolia* Scop.) have been observed [13]. In addition, the application of gamma irradiation on barley seeds enhanced root and sprout development while concomitantly stimulating the antioxidant defense [14]. Another very promising technique in seed physical pretreatment is the exposure of seeds to static magnetic field (SMF). Seed treatments with SMF have been shown to improve germination rates, plant growth, and crop yield. As such, cucumber seeds (*Cucumis sativus* L.) were exposed to SMF in a range from 100 to 250 mT for multiple hours and increased the germination percentage (19%), rate of germination (49%), length of seedling (34%), and dry weight (33%) in magnetoprimed seeds compared to unexposed seeds [15]. The effect of different SMF intensities (20, 42, 125, and 250 mT) on barley seeds was investigated and showed that lower intensities of SMF treatment (≤125 mT) enhanced seed germination [16]. These positive effects on biometrical parameters can be attributed to the beneficial impact of SMF on various biological processes. For example, SMF treatment of flax seeds (*Linum usitatissimum* L.) changed physiological processes, such as respiration, photosynthesis, nutrient uptake, water balance, and biochemical characteristics, including genes involved in redox homeostasis as well as enzymes, proteins, and secondary metabolites [17]. In addition, germinating sunflower seeds (*Helianthus annuus* L.) treated with SMF exhibited significantly higher enzymatic activities of α-amylase, dehydrogenase, and protease compared to untreated controls, alongside an improved seed coat membrane integrity [18]. Additionally, SMF treatments increased catalase activity and proline content in wheat seedlings (*Triticum aestivum* L.) while reducing the activity of peroxidase, the rate of lipid peroxidation, and electrolyte leakages from the seeds [19]. These findings highlight the beneficial effects of SMFs on growth characteristics as well as on biochemical and physiological properties. However, the duration and strength of SMF are important factors to take into account for optimal conditions [19,20].

Seed priming is another strategy to improve germination and involves the controlled hydration of seeds prior to sowing. It can further be classified based on the priming components used, more specifically hydropriming, osmopriming, hormopriming, nutripriming, and biopriming [8]. The latter involves controlled hydration of seeds along with the inoculation of biological agents (beneficial microorganisms). Common techniques for incorporating beneficial microorganisms, either individually or as consortia, encompass seed coating, seed soaking, or immersing seedlings in a microbial suspension to establish beneficial microorganisms within the plant microbiome [7]. Among these microorganisms are cyanobacteria, which have been the focus of several studies exploring their extracts and potential applications to plants. As such, it has been shown that cyanobacteria have the ability to increase the vigor index, root/shoot length, and weight, as well as crop yield [21]. In addition, cyanobacterial inoculation can improve rice seed (*Oryza sativa* L.) germination and growth parameters [22]. According to Osman et al., the amount of growth-promoting secondary metabolites varies depending on the cyanobacterial strain used [23].

Research on the combined effect of SMF and biopriming is at present limited. Jovičić-Petrović demonstrated that the combined application of SMF and biopriming with a *Bacillus* bacterial strain improved old seed revitalization and seed germination [24]. However, research on the impact of SMF on a cyanobacterial consortium prior to the biostimulant extraction and its application as a biopriming agent has, to our knowledge, not yet been investigated. Therefore, the objective of the current study was to investigate the effects of biopriming of *Cucumis sativus* L. seeds with an extract derived from a consortium of nitro-fixing cyanobacteria. These consortia were grown with or without prior exposure to SMF treatment before the biostimulant extract was prepared. The impact of biopriming was assessed by evaluating various biometrical parameters related to germination and seedling growth of *Cucumis sativus* L. var. Marketmore as well as elemental concentrations in plants The experiments were carried out with *C.ucumis sativus* L., a horticultural crop, and the third most produced vegetable developed as a new model plant [25]. To the best of our knowledge, this is the first study to explore the effects of SMF treatment on cyanobacterial consortium inocula prior to their growth and subsequent application in seed biopriming.

## 2. Results

### 2.1. Auxin Concentration in the Biomass of the Nitrofixing Cyanobacteria Consortium with Static Magnetic Field

Tryptophane and three auxin derivates were measured in the nitrofixing cyanobacterial biomass and are shown in Table 1. The results only show significant differences (*p* < 0.05) between the concentrations of indole-3-butyric acid (IBA) between the C2-1 and C2-3 consortium on the one hand and the C2-2 consortium on the other hand, which displayed a lower concentration.

### 2.2. Germination Parameters

The results showed no statistically significant differences in the germination parameters, i.e., germination percentage [26], mean germination time (MGT), germination rate (GR), and germination speed coefficient (GSC), between the biopriming treatments of *Cucumis sativus* L. seeds. However, there was a statistically significant difference with regard to the vigor index I. The control and consortium C2-3 exposed plants had the lowest vigor index I, and plants bioprimed with extracts from consortia C2-1 and C2-2 displayed the highest vigor index I (Table 2). The biopriming of an extract of nitrofixing cyanobacteria of consortium C2-1, without SMF treatment, or C2-2, with an SMF induction between 40–50 mT, promoted the Vigor index I with 67% and 65%, respectively.

In addition, the root and stem lengths were measured 7 days after germination (Figure 1). Statistically significant differences in root length were observed between the control and C2-3 *Cucumis* seedlings on the one hand and the seedlings that received biopriming from C2-2 and C2-1 consortia. A significant increase in root length was observed, with C2-1 showing a 137% increase compared to the control and C2-2 demonstrating a 117% increase.

For the stem length, all bioprimed seedlings were taller than the control plants. The increase in hypocotyl length was 34% in C2-1 and C2-2 bioprimed plants and 25% for C2-3 bioprimed plants with respect to the control plants.

### 2.3. Growth Parameters of Cucumis sativus L. Seedlings at 15 Days

In the second bioassay, root and stem measurements were made 15 days after seedling growth. Root length increased significantly in bioprimed seedlings compared to the control. In addition, biopriming with C2-1 and C2-2 outperformed C2-3 bioprimed plants with respect to the root length (Figure 2). A significant increase of 144% for root length was observed in C2-2 bioprimed seedlings, where as this was a 110% or 77% increase in C2-1 or C2-3 bioprimed seedlings as compared to the controls. Stem length increased significantly in seedlings bioprimed with C2-2 compared to the other treatments. Specifically, C2-2 bioprimed seedlings showed a 65% increase, followed by 25% in C2-1 bioprimed seedlings and 12% in C2-3 bioprimed seedlings (not significant), all in comparison to the control plants.

The fresh root and leaf weight of *Cucumis sativus* L. var. Marketmore was determined 15 days after germination with biopriming with a consortium of nitrofixing cyanobacteria at different ranges of magnetic induction (Figure 3a,b) or no biopriming (control). Root fresh weight increased significantly in seedlings bioprimed with the extract from C2-2 compared to the other bioprimed treatments that were also higher than the control plants (Figure 3a). Leaf fresh weight increased significantly in all bioprimed seedlings compared to the control seedlings (Figure 3b). Nevertheless, the ratio of fresh weight to dry weight of leaves remained similar between the different treatments (Figure 3c).

### 2.4. Macroelements and Microelements in the Leaves of Cucumis sativus L., var. Marketmore

In addition, the concentrations of macro and microelements in the leaves of *Cucumis sativus* L. var. Marketmore at 15 days of germination was determined (Table 3). The concentrations of the macroelements P and S were significantly higher in the control plants compared to the other treatments. Also, the concentrations of the microelements Cu, Al, Na, and Zn were significantly higher in the control plants as compared to plants from the other treatments.

## 3. Discussion

The urgent need to increase agricultural productivity while mitigating the environmental impacts of conventional farming practices has boosted research in alternative techniques that enhance seed germination and early plant growth. Climate-related stressors, such as higher temperatures and water scarcity, further highlight the necessity for resilient crop establishment and enhanced germination rates to support food security. Seed priming through SMF applications or biopriming has shown promising results. This study contributes to this emerging field by exploring the combined effects of SMF exposure to cyanobacterial consortia and seed biopriming of *Cucumis sativus* L. seeds with these extracts on seed germination and early seedling development. The findings provide insights into optimizing these treatments to enhance agricultural resilience in changing and challenging environmental conditions.

Biopriming stands out as an eco-friendly technique for increasing the rate of seed germination and overall fitness of plants by playing an important role in various events in the plant’s life cycle [8]. Improvement in growth due to seed priming could be due to the release of plant growth-regulating compounds by the cyanobacteria. Paul and Nair (2008) [27] reported stimulation of plant growth due to plant growth regulating hormones and compounds such as amino acids, sugars, and vitamins excreted by cyanobacteria. The secretion of these compounds not only promoted growth but mitigated the adverse impact due to abiotic stress in plants [28]. The growth of plants used in dry land restoration improved due to seed biopriming with cyanobacteria [29]. Osman et al. (2021) [23] observed a positive impact on the growth of wheat plants due to priming of the seeds with cyanobacterial extracts.

Cyanobacteria can increase root and stem growth, dry weight, and wheat yield [30,31]. Cyanobacterial extracts improved nutrient uptake and plant development in lettuce (*Lactuca sativa* L.), tomato (*Solanum lycopersicum* L.) [32,33], and cucumber [34]. In a broader sense, cyanobacteria are used as commercial bioinoculants to promote plant development due to their higher biodiversity, ability to survive in various conditions, faster growth rate, and simpler nutritional requirements. The results obtained in the present study confirm those reported previously by these authors.

In general, the results of our study show that SMF treatments of nitrofixing cyanobacterial consortia can have beneficial effects on their extracts to be used for biopriming. Nevertheless, it is clear that these are dependent on the ranges of magnetic induction and can be different at different stages of the life cycle in *Cucumis sativus* L. var. Marketmore. As such, it is clear that based on the Vigor index I, extracts from consortia C2-1 (no SMF treatment) or C2-2 (low SMF treatment) have a positive effect on germination. This is a measure to evaluate the health and strength of seedlings during the early stages of growth and is an important predictive value for the potential of seeds to develop into strong plants under field conditions [7]. Reports on the influence of cyanobacterial biopriming on the vigor I index have already been reported. Studies of cyanobacteria using different concentrations and different inoculant compositions have shown positive effects on germination. As such, biopriming with indigenous cyanobacteria stimulated root length and showed the potential for improving native plant establishment in degraded soils, particularly in challenging restoration environments like those in mining areas [29]. Also, applying *Arthrospira maxima* on basil (*Ocimum basilicum* L.) stimulated seedling growth, but clear differences were obtained between different concentrations [35]. This suggests that a concentration range of our inoculant should be tested to optimize the application concentration based on the positive effects of germination parameters.

Besides biopriming, SMF treatment has also demonstrated significant potential in enhancing plant growth and development within agricultural practices. Magnetic field treatment of sunflower seeds has been shown to improve plant growth and yield [36], and the use of MF in the range of 2.23–3.72 mT had a positive impact on the germination time of triticale seeds [37]. An increase in the germination rate and seedling length was also observed after the pretreatment of radish (*Raphanus sativus* L.) seeds with the use of an SMF at different inductions (8 mT and 20 mT) [38]. With the results obtained for root length, fresh root weight, and stem length, it can be observed that the biopriming with the extract from the consortium of nitrofixing cyanobacteria treated with 40–50 mT magnetic induction has a positive effect on the *Cucumis sativus* L. seedling growth (Table 2, Figure 1, Figure 2 and Figure 3). Therefore, it can be confirmed that SMF application to the cyanobacterial consortium enhances their properties and might positively influence these parameters. Nevertheless, it is also clear that the application of different SMF inductions has a different effect on these properties and, hence, their outcome as a priming agent.

Due to the large amount of bioactive compounds that cyanobacteria possess, they can influence the physiology of plants, being reflected in higher biomass production [39,40]. This is in agreement with the results obtained in this study since the use of the consortium with the different magnetic inductions had a greater influence than the control, improving the fresh weight of the root as well as the root length and fresh weight of the leaves of *Cucumis sativus* L. (Figure 1, Figure 2 and Figure 3).

The positive effect that both cyanobacteria and the SMF have on plants is well known, but there have been only a few studies that apply both treatments. Moreover, this study aims to investigate SMF treatment of the consortia during their growth prior to their application in biopriming. In order to further optimize this technology, it is important to gain better insight into the mode of action of these treatments on plant growth and development. Seed metabolic activities and germination-related parameters are used as appropriate indices to understand the effects of cyanobacteria on seed physiology to generate a healthy crop. On the other hand, many researchers indicate that the application of MF causes changes in biochemical and physiological processes in seeds and plants, such as enhanced photosynthesis efficiency potentially through improved electron transport and increased antioxidant activities, which have a positive effect on the health and productivity of plants [41]. A potential factor playing an important role in plant growth is the contribution of plant hormones, which can be provided by the consortium. Cyanobacteria actively promote seed germination, plant growth, and development due to their ability to produce some plant hormones, such as auxins, cytokinins, and gibberellins, from the genera *Anabaena*, *Anabaenopsis*, and *Calothrix* [42]. Indole-3-acetic acid [43] has previously been identified as a product of cyanobacteria, with various benefits provided to plants. A strong positive correlation was found between the IAA produced by cyanobacteria and chamomile (*Matricaria chamomilla* L.) growth metrics [44]. However, the IAA concentration determines whether the effects are positive or negative, with many studies showing that excessive amounts of IAA can inhibit growth, have negative effects on plant physiology, and potentially promote seed dormancy [45]. Auxins promote stem elongation at concentrations of 10^−6^–10^−5^ M, while at these same concentrations, they inhibit root growth [46]. Positive plant responses are, therefore, concentration-dependent and controlled by the spatiotemporal distribution of IAA [47], meaning that optimal levels may vary between plant species, and as a result of the plant’s tissue sensitivity to auxins [45]. Although there were not many significant differences observed in auxin derivate concentrations between our consortia (Table 1), it is clear that the amount of tryptophan in consortium C2-2 is the highest and Indole-3-butyric acid (IBA) the lowest. This might, therefore, partially explain the subsequent plant growth differences observed in the *Cucumis* seedlings. External application of tryptophan (in our case, derived from the consortium C2-2) might enhance crop productivity by promoting endogenous auxin biosynthesis of the plants, which further regulates key growth and development processes. It might boost nutrient uptake and plant resilience under stress, significantly improving yield and growth [48].

In addition, the consortium of nitrofixing cyanobacteria treated with the different magnetic inductions stimulated plant growth, causing a dilution of nutritional elements (micro-macro element concentrations; Table 3). However, the plants did not suffer from deficiencies because the FW/DW ratios were maintained (Figure 3c), which implies that the plants maintained adequate water content and balanced growth, critical indicators of healthy development. As plant growth progresses, especially in young seedlings, a decline in P concentrations is often observed, likely due to a dilution effect, where newly accumulated biomass distributes available P over a larger plant size. This pattern aligns with our observations of the production of carbon-rich structural compounds, supporting expanding tissues, while nutrient concentrations, including P, decrease as a result of resource allocation dynamics [49]. The nitrofixing cyanobacteria extract used in the biopriming might stimulate the uptake of P, which is used in the photosynthetic process and energy production, further stimulating seedling growth, an effect that is less pronounced in the control plant [8,46]. Cyanobacteria can improve the overall nutrient uptake efficiency of plants, leading to better growth and yield [50]. However, this might not always translate to higher nutrient concentrations in leaves [51], but due to their increased growth, a higher total plant nutritional content is expected. The content of macro and microelements are important determinants of plant growth and plant quality. The highest concentrations of P, Mg, Fe, Mn, and Zn are found in the leaves and are related to the fact that the leaf is the organ with the highest photosynthetic activity, unlike the fruits [52]. It has been shown that minerals are found in a higher concentration in the leaf because they are the main organs where nutrients are distributed after having been absorbed by the root, which in turn guarantees the flow of nutrients in the entire plant [53]. Cyanobacteria-based formulations have increased Fe, Zn, Mn, and Cu soil contents [54] as well as micronutrient concentrations in plant parts, such as Zn in maize leaves [55] and Fe, Zn, Mn, and Cu in wheat grains [56]. The mechanisms of plant micronutrient enrichment by soil microalgal inoculations are not well understood, but one proposed mechanism is the production of siderophores by cyanobacteria [57]. In our study, we used extracts from cyanobacterial consortia in which siderophore production was no longer present. This absence might explain how compounds such as tryptophan can stimulate plant growth through hormonal pathways while supporting healthy development without causing nutrient deficiencies. Plant micronutrient deficiency leads to susceptibility to diseases in food crops, so elucidating the potential role of microalgae in addressing this issue is of critical value [55,56,58].

The extract of nitrofixing cyanobacteria consortium with the different treatments showed their biostimulant effect on *Cucumis sativus* seeds. However, the mechanism of action of these extracts, either through the action of phytohormones, other beneficial substances, or the synergistic effect of both, is still unclear. Multi or metaomics analyses may help to assess the molecular mechanisms of plant cyanobacterial consortium interactions. In addition, our approach ensures that seedlings are first cultivated under controlled conditions, allowing them to establish and gain strength before being transplanted into open-air environments. This initial phase of controlled growth enhances their resilience and adaptability, improving their chances of successful establishment once exposed to external conditions. However, future field scale studies to follow-up plant growth and productivity in the field are important to assess this contribution to sustainable agriculture.

In conclusion, the biostimulant potential of the nitrofixing cyanobacteria consortia is verified with different ranges of magnetic induction in the germination of *Cucumis sativus* L. var. Marketmore seeds, which would support its agricultural applications. In this regard, future research should focus on the underlying mode of action of the extract of the C2-2 consortium, as it demonstrated the highest plant growth-stimulating activity. This consortium, treated with magnetic induction in the range of 40–50 mT, presents the best performance in key seedling developmental parameters, including of stem and root lengths, vigor index I as well as root and leaf fresh weights, highlighting its potential in biopriming applications. Nevertheless, all biopriming treatments demonstrated enhanced plant growth compared to the control, highlighting the need to explore optimal conditions for this agricultural practice. Such optimization holds the potential not only to boost crop production but also to improve plant nutritional value, an essential factor for human health and consumption.

## 4. Materials and Methods

### 4.1. Extract Preparation

The inoculum consisting of a natural consortium identified and isolated from the rhizosphere of *Carica papaya* made up of *Nostoc commune*, *Calothrix* sp., and *Aphanothece minutissima* (C2) was exposed to SMF for 1 h [59]. The static magnetic fields (SMF) were generated by a ferrite magnetic device. The magnetic intensity was 40–50 mT and 100–200 mT, which was measured by a Gaussmeter measuring device (CNEA, Santiago de Cuba, Cuba).

The static magnetic fields (SMF) were generated by a ferrite magnetic device. The magnetic intensity was 40–50 mT and 100–200 mT, which was measured by a Gaussmeter measuring device (CNEA, Cuba). The magnetic field measurement procedure was carried out by using a Hall effect Teslameter (Lakeshore Model 475-DSP, Series 42302, Westeville, OH, USA) equipped with the Sensor (Model HMMT-6J04-VR, Series HA2580, Westeville, OH, USA), which has been calibrated against the reference of the magnetic induction standard found in CNEA. The measurement procedure consists of measuring the transverse component (perpendicular to the axis) of the magnetic induction along the axis, from one end to the other of the working region, with a constant step that allows obtaining no less than 10 points, using a transverse hall probe. A system will be used for the millimetric positioning of the hall probe with three degrees of freedom in mutually perpendicular directions (X, Y, and Z axes). The results determine the average value of the magnetic induction in the work region and the standard deviation. The glass bottles with samples were placed in the center of the magnetic device to expose the cultures to the SMF, guaranteeing the same intensity of light and aeration as the control cultures. The intensity of the SMF applied to the cultures was chosen following the results obtained in previous research [59].

The SMF was applied to a 100 mL inoculum for 1 h before being inoculated in one liter of BG11_0_ culture medium. The control culture (C2-Control) was maintained under the same conditions (temperature, light, aeration, and nutrients). All assays were performed in triplicate. Two treatments were given: an SMF of 40–50 mT induction (treatment C2-2) and a second group with 100–200 mT induction (C2-3). A control group (C2-1) was established and did not receive magnetic treatment.

The temperature conditions in the culture lab were stable at 22 ± 2.3 °C. An LED light panel was used to keep the cultures illuminated under continuous light. The light intensity was adjusted daily with a traceable light meter, maintaining the cultures with an illumination of 48.83 μE m^−2^s^−1^. Cultures were aerated by bubbling filtered air through 0.20 µm glass microfiber syringe filters. The device to aerate the cultures was an air pump for aquariums and ponds (SERA air 550 R plus, Heinsberg, Germany) with an airflow capacity of 9.2 Lmin^−1^. The SMF was applied in a 100 mL inoculum for one hour before being inoculated in one liter of BG11_0_ culture medium [60]. At the end of 21 days of culture, each sample was freeze-dried, and the auxin concentrations were measured in the lyophilized biomass in collaboration with Prof. E. Prinsen from the University of Antwerp, Belgium.

Cyanobacterial extracts were prepared at a concentration of 10 mg mL^−1^, 300 mg of freeze-dried biomass in 30 mL of distilled water (dH_2_O) and kept for 12 h under constant stirring at 300 rpm. It was then heated at 50 °C for 1 h, with the same stirring intensity [61]. The mixture was allowed to cool down to room temperature and then sonicated for 5 min. Afterward, it was centrifuged at 4500 rpm for 5 min, and the supernatant was filtered with a sterile filter. In the end, three cyanobacterial extracts were obtained.

### 4.2. Auxin Analysis

To determine the auxin concentration and auxin biosynthetic precursors, 5 mg of freeze-dried biomass was taken from the different treatments. Hormones were separated by a UPLC-MS/MS system. Full details are provided in De Paepe et al. (2024) [62].

### 4.3. Biopriming of Cucumis sativus L. var Marketmore Seeds

For the biopriming experiments, certified seeds (OSRO/CUB/01/CHA) of *Cucumis sativus* L. var Marketmore were used. Prior to germination, 100 seeds were washed for 15 min with 4% sodium hypochlorite and then rinsed three times with Milli-Q H_2_O. Subsequently, they were placed in a 0.1% HCl solution on a sieve (WiseShake SHO-2D, Wertheim, Germany). Next, three successive washes were carried out with dH_2_O, immersed in 70% alcohol for 5 min under a hood, and washed again three times with Milli-Q water.

For each treatment, 100 seeds were imbibed in the three cyanobacterial extracts for 1 h or dH_2_O for the control condition. Then, 25 seeds were placed equidistantly in each sterile Petri dish (9 cm diameter) with filter paper (HAO HF) with the addition of 3 mL of cyanobacterial extract or dH_2_O (control). Four replicates per treatment for a total of 100 seeds. Seeds were incubated at a temperature of 30 °C and a light intensity of 58.59 µE m^2^s^−1^ for 7 days).

### 4.4. Germination Parameters

The germinated seeds were counted, and the length of the stem and radicle of each germinated seedling was measured using a ruler. Germination percentage [26], mean germination time (MGT), germination rate (GR), germination speed coefficient (GSC), and vigor index I were also evaluated. The respective formula for each parameter is:

Germination percentage [26] is the proportion of seeds germinated expressed as a percentage of the total seeds tested [63].GP (%) = (G/N) × 100 (1)
in which G stands for the total number of seeds that were germinated on the seventh day and N for the total number of seeds sown

Mean germination time (MGT) is also known as resistance to germination or the inverse of the coefficient of speed. It measures the average germination time needed for the seeds to germinate. The values are expressed in days, according to Kader 2005 [64]MGT = ∑(Ni × di)/N (2)
where Ni is the percentage of seeds germinating on the ith day, and ti is the number of days counted from the start of the experiment (i) to k, the last day on which seeds germinated.

Germination speed coefficient (GSC) gives an indication of the rapidity of germination. It increases when the number of germinated seeds increases, and the time required for this germination decreases. Theoretically, the highest GSC possible is 100. This would occur if all seeds germinated on the first day. This value is calculated with the formula described by Kader 2005 [64].GSC (% day^−1^) = N1 + N2 + … NX/100 × (N1T1 + … + NXTX) (3)
where N_i_ is the number of seeds germinated each day (first day = 1; final day = X), and T_i_ is the number of days for seedlings corresponding to N_i_.

Stem and root length were measured from the exit of the seed hypocotyl to where the epicotyl begins (stem length) and root length. A graduated ruler (cm) was used according to the [65] methodology.

Vigor index I is measured by taking the length of the root and shoot of each individual seedling and the GP into account [66].Vigor Index I = GP × Seedling length (radicle and stem) (4)

### 4.5. Seedling Growth

A seedling growth assay was carried out after seven days of evaluating the germination parameters. Twenty-five seedlings in good phytosanitary condition and homogeneous growth were planted in small plastic containers (100 mL volume) filled with loamy sand soil collected in Ravels (Belgium) [67]. Plants were kept in the growth chamber with a 12 h/12 h day/night photoperiod, a day/night temperature of 22/18 °C, and a relative humidity of 65%, to continue their growth until 15 days. Then, the stem, root length, and fresh weight were assessed.

### 4.6. Elemental Determination of Seedlings

The elemental concentrations of magnesium (Mg), potassium (K), phosphorus (P), calcium (Ca), copper (Cu), iron (Fe), aluminum (Al), manganese (Mn), sodium (Na), sulfur (S), and zinc [68] were measured in leaves of 15-day-old seedlings. The samples were dried in an oven at 70 °C until they were completely dry, pulverized, and approximately 200 mg was transferred to open heat-resistant tubes (SCHOTT DURAN^®^, Wertheim, Germany) followed by the addition of 1 mL 69% HNO_3_ (ARISTAR^®^ for trace analysis, Częstochowa, Poland) and left overnight at room temperature. On the next day, the tubes were transferred to heating blocks and the temperature was slowly increased to 110 °C and kept there until the tubes were completely evaporated. This step of adding 1 mL 69% HNO_3_ was repeated three times. The last digestion step included the addition of 1 mL of 37% HCl (ARISTAR^®^ for trace analysis), and finally, the samples were dissolved in a total volume of 5 mL of 2% HCl (diluted with milliQ H_2_O). The concentration of the elements was measured through inductively coupled plasma-optical emission spectrometry (ICP-OES 710, Agilent Technologies, Santa Clara, CA, USA).

### 4.7. Data Processing and Statistical Analysis

A complete randomized design was applied. The data obtained were subjected to Kolmogorov–Smirnov normality test and homoscedasticity, and the analysis of variance (one-way ANOVA) was performed. Fisher’s least significant difference (LSD) multiple comparison test was carried out with a significance level of *p* < 0.05 using the statistical program Statgraphics version 19—X64 (Company Statgraphics Centurion, Virginia, VA, US).

## Figures and Tables

**Figure 1 plants-14-00628-f001:**
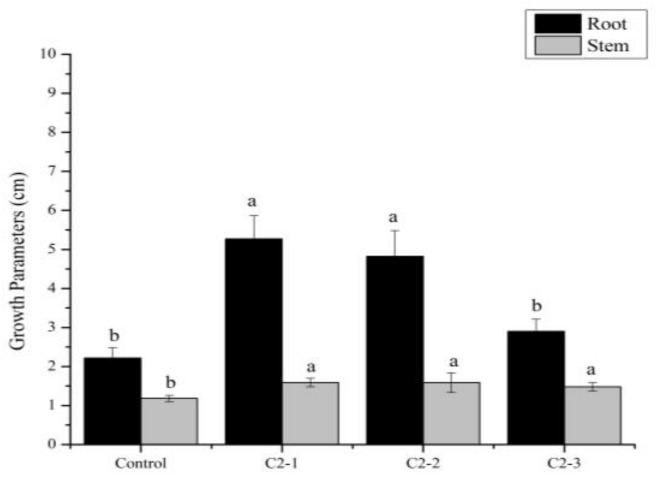
Growth parameters (stem and root length) in *Cucumis sativus* L. var. Marketmore seeds 7 days after germination on a plate, without biopriming (H_2_O: control) or with biopriming (C2-1: consortium not exposed to SMF; C2-2: consortium exposed to SMF of 40–50 mT; C2-3: consortium exposed to SMF of 100–200 mT). Mean ± SE with different letters indicates statistically significant differences. (LSD test) (*p* < 0.05).

**Figure 2 plants-14-00628-f002:**
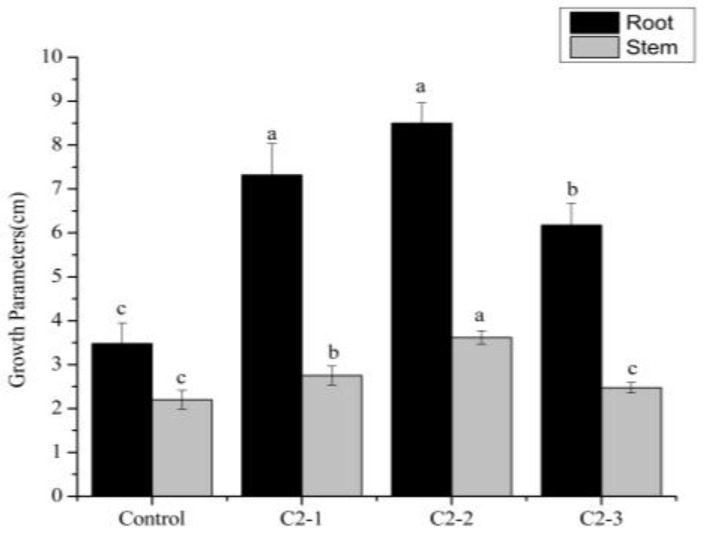
Growth parameters (stem and root length) in *Cucumis sativus* L. var. Marketmore seeds 15 days after germination on soil, without biopriming (H_2_O: control) or with biopriming (C2-1: consortium not exposed to SMF; C2-2: consortium exposed to SMF of 40–50 mT; C2-3: consortium exposed to SMF of 100–200 mT). Mean ± SE with different letters indicates statistically significant differences. (LSD test) (*p* < 0.05).

**Figure 3 plants-14-00628-f003:**
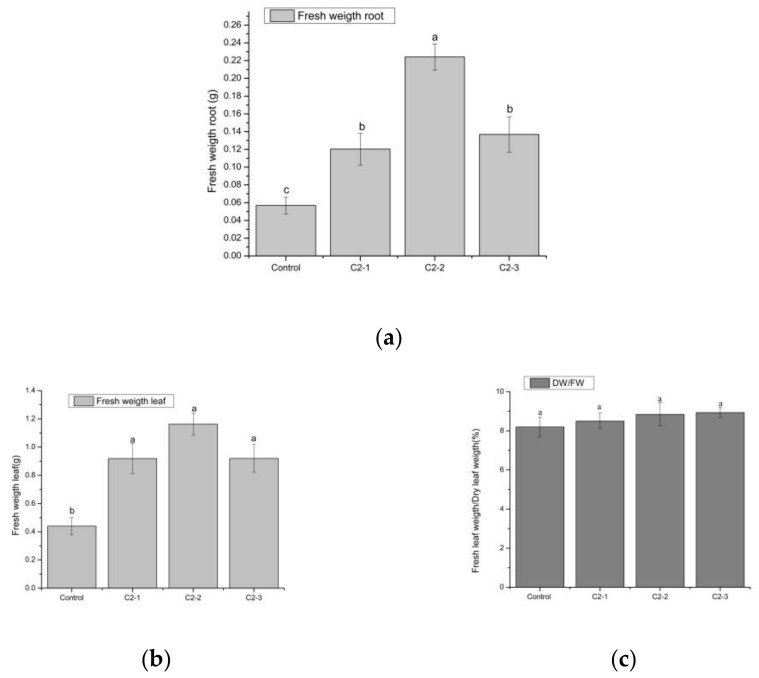
(**a**) Root fresh weight and (**b**) leaf fresh weight of *Cucumis sativus* L. var. Marketmore at 15 days of germination without biopriming (only H_2_O: control) or with biopriming (C2-1: consortium not exposed to SMF; C2-2: consortium exposed to SMF of 40–50 mT; C2-3: consortium exposed to SMF of 100–200 mT). (**c**) Fresh weight (FW)/dry weight (DW) ratio of *Cucumis sativus* L. leaf at 15 days of germination. Mean ± SE (n = 4) with different letters indicate statistically significant differences. (LSD test) (*p* < 0.05).

**Table 1 plants-14-00628-t001:** Auxin concentration (pmol/mg DW) in the consortium of nitrofixing cyanobacterial biomass with SMF. Values represent the average ± SE of 3 biological replicates.

Parameters	C2-1	C2-2	C2-3
Indole-3-aldehyde	3.95 ± 1.30 ^a^	2.38 ± 0.18 ^a^	2.75 ± 0.65 ^a^
Indole-3-acetic acid	1.98 ± 0.56 ^a^	1.75 ± 0.33 ^a^	3.29 ± 0.88 ^a^
Tryptophan	1404 ± 530 ^a^	3085 ± 1092 ^a^	2111 ± 663 ^a^
Indole-3-butyric acid	981 ± 96 ^a^	512 ± 45 ^b^	1183 ± 73 ^a^

Treatment: C2-1 = consortia without SMF treatment; C2-2 = consortia with SMF treatment in the range of 40–50 mT; C2-3 = consortia with SMF treatment in the range of 100–200 mT. Mean ± SE with different letters indicates statistically significant differences (LSD test) (*p* < 0.05).

**Table 2 plants-14-00628-t002:** Germination parameters evaluated in the germination of *Cucumis sativus* L. var. Marketmore. Values represent the average ±SE of 100 seeds.

Parameters	Control	C2-1	C2-2	C2-3
Germination percentage (%)	65± 8 ^a^	54 ± 8 ^a^	57± 4 ^a^	57± 4 ^a^
Mean germination time [9]	3.09 ± 0.16 ^a^	3.22 ± 0.30 ^a^	3.43 ± 0.16 ^a^	3.06 ± 0.32 ^a^
Germination rate (seed day^−1^)	2.30 ±0.30 ^a^	1.93 ± 0.30 ^a^	2.04 ± 0.16 ^a^	2.04 ± 0.17 ^a^
Germination speed coefficient (% day^−1^)	32.67 ± 1.71 ^a^	31.82 ± 2.87 ^a^	30.61 ± 1.41 ^a^	33.83 ± 3.57 ^a^
Vigor index I	221 ±19 ^b^	370 ± 37 ^a^	365 ± 18 ^a^	250 ± 21 ^b^

Without biopriming (only H-O: control) or with biopriming of a consortium of nitrofixing cyanobacteria at different ranges of magnetic induction (C2-1: consortium not exposed to SMF; C2-2: consortium exposed to SMF of 40–50 mT; C2-3: consortium exposed to SMF of 100–200 mT). Mean ± SE with different letters indicates statistically significant differences. (LSD test) (*p* < 0.05).

**Table 3 plants-14-00628-t003:** The concentration of minerals in the leaves of *Cucumis sativus* L. (mg/kg DW) at 15 days. Values represent the average ± SE of 4 biological replicates.

Macroelements (mg kg^−1^ DW)				
Elements	Control	C2-1	C2-2	C2-3
Mg	5009 ± 563 ^a^	3898 ± 197 ^a^	4282 ± 500 ^a^	3841 ± 7 ^a^
K	17,416 ± 3582 ^a^	18,348 ± 1027 ^a^	19,601 ± 2799 ^a^	17,435 ± 425 ^a^
P	14,757 ± 2356 ^a^	6538 ± 269 ^b^	6569 ± 517 ^b^	8093 ± 279 ^b^
Ca	19,493 ± 3679 ^a^	24,030 ± 1672 ^a^	24,987 ± 3983 ^a^	26,074 ± 2979 ^a^
S	5178 ± 218 ^a^	4237 ± 288 ^b^	3937 ± 317 ^b^	4372 ± 221 ^b^
**Microelements (mg kg^−1^ DW)**				
Cu	13.38 ± 2.02 ^a^	9.71 ± 0.53 ^b^	9.46 ± 0.96 ^b^	10.09 ± 0.61 ^a^
Fe	188 ± 23 ^a^	125 ± 14 ^a^	146 ± 29 ^a^	121 ± 8 ^a^
Al	211 ± 34 ^a^	82 ± 15 ^b^	92 ± 26 ^b^	106 ± 10 ^b^
Mn	61 ± 8 ^a^	68 ± 6 ^a^	84 ± 11 ^a^	72 ± 3 ^a^
Na	4157 ± 379 ^a^	2394 ± 137 ^b^	1613 ± 213 ^c^	2785 ± 196 ^b^
Zn	71 ± 13 ^a^	45 ± 2 ^b^	52 ± 6 ^b^	54 ± 3 ^b^

Without biopriming (only H_2_O: control) or with biopriming (C2-1: consortium not exposed to SMF; C2-2: consortium exposed to SMF of 40–50 mT; C2-3: consortium exposed to SMF of 100–200 mT). Mean ± SE with different letters indicates statistically significant differences. (LSD test) (*p* < 0.05).

## Data Availability

The original contributions presented in this study are included in the article. Further inquiries can be directed to the corresponding authors.

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
