# Peer review of "Biopriming of *Cucumis sativus* L. Seeds with a Consortium of Nitrofixing Cyanobacteria Treated with Static Magnetic Field"

_plants, 2025, doi:10.3390/plants14040628_

Round 1
Reviewer 1 Report
Comments and Suggestions for Authors
Some editing (spacing, italics, etc.) is needed. The comments point out the content suggestions.

Author Response
Comments 1: Change in abstract line 29 sustainable agricultural to (Horticultural crop because Cucumis sativus is a horticultural crop)
Response 1: Thank you for pointing this out. We agree with this comment. We have, therefore, changed this to “for horticultural crops” in the abstract on page 1, line 29.
Comments 2: Change in line 117 to study previously (rephrase)
Response 2: We adjusted the sentence: change on page 3, paragraph 3, line 116-117: “Jovičić‐Petrović demonstrated that the combined application of SMF and biopriming…”
Comment 3: Italic. Maybe will be valuable to state why choosing Cucumis for de experiment (in regard with the species germination and growth without treatment.
Response 3: We agree and have, accordingly modified the introduction on page 3, paragraph 3, lines 127-128.
“The experiments were carried out with Cucumis sativus L., a horticultural crop and the third most produced vegetable developed as a new model plant [1].”
Comments 4: Move this explanation as a footer of tables.
Response 4: We agree with the reviewer. It was made and moved to page 4, Table 1, lines 146 and 147.
Comments 5: It would be useful to argue the higher concentrations of micro and macro elements in the control plants as shown in the results.
Response 5: We agree with the reviewer that we have to discuss these finding more extensively and therefor made changes on page 9, lines 346-354. “As plant growth progresses, especially in young seedlings, a decline in P concentrations is often observed, likely due to a dilution effect, where newly accumulated biomass distributes available P over a larger plant size. This pattern aligns with our observations of the production of carbon-rich structural compounds, supporting expanding tissues, while nutrient concentrations, including P, decrease as a result of resource allocation dynamics [2]. The nitrofixing cyanobacteria extract used in the biopriming might stimulate the uptake of P, which is used in the photosynthetic process and energy production, further stimulating seedling growth, an effect that is less pronounced in the control plant [3,4].”
- Janicka, M.; Reda, M.; Napieraj, N.; Michalak, A.; Jakubowska, D.; Kabała, K. Involvement of diamine oxidase in modification of plasma membrane proton pump activity in Cucumis sativus L. seedlings under cadmium stress. International Journal of Molecular Sciences 2022, 24, 262, doi: https://doi.org/10.3390/ijms24010262.
- Ouyang, M.; Tian, D.; Niklas, K.J.; Yan, Z.; Han, W.; Yu, Q.; Chen, G.; Ji, C.; Tang, Z.; Fang, J. The scaling of elemental stoichiometry and growth rate over the course of bamboo ontogeny. New Phytologist 2024, 241, 1088-1099, doi: https://doi.org/10.1111/nph.19408.
- Srivastava, S.; Tyagi, R.; Sharma, S. Seed biopriming as a promising approach for stress tolerance and enhancement of crop productivity: A review. Journal of the Science of Food Agriculture 2024, 104, 1244-1257, doi:https://doi.org/10.1002/jsfa.13048.
- Taiz, L.; Zeiger, E. Photosynthesis: physiological and ecological considerations. 2002, 9, 172-174, doi:https://doi:10.1093/aob/mcg079.

Reviewer 2 Report
Comments and Suggestions for Authors
The study introduces a seed biopriming method that applies static magnetic fields (SMF) to nitro fixing cyanobacterial consortium for Cucumis sativus L. seeds. The technique brings beneficial outcomes to sustainable agriculture which is currently needed for modern agricultural practice. However, a few comments aim to provide constructive feedback that could enhance the quality and impact of this research paper presented.
1. It would be valuable, to add a detailed description of the methodology used for the SMF treatment and the preparation of cyanobacterial extracts.
2. Could the authors provide more detailed information on the specific parameters used during the SMF treatment, such as duration and exact field strength?
3. It would be beneficial for the authors to discuss how environmental factors might influence the effectiveness of biopriming with cyanobacteria and SMF.
4. For the reader's interest, the presentation of data, particularly in figures and tables, should be improved for clarity. Please ensure that all figures are clearly labeled and that legends provide sufficient context.
5. To strengthen the argument for the significance of findings and provide a broader context for their implications in agricultural practices. The discussion section could be expanded to include a more comprehensive analysis of how the observed effects of biopriming and SMF treatment relate to existing literature.
6. It would be beneficial to outline specific experiments or approaches that could be taken to investigate these mechanisms.
7. It would be beneficial to discuss how environmental factors, such as soil type and climate conditions, might influence the effectiveness of biopriming with cyanobacteria and SMF.
Author Response
Comments 1: It would be valuable, to add a detailed description of the methodology used for the SMF treatment and the preparation of cyanobacterial extracts.
Response 1: Thank you, we agree with this comment. Changed on page 10, paragraph 5 and page 11, paragraph 1, lines 407- 421. “The static magnetic fields (SMF) were generated by a ferrite magnetic device. The magnetic intensity was 40-50 mT and 100-200 mT, which was measured by a Gaussmeter measuring device (CNEA, Cuba). The magnetic field measurement procedure was carried out by using a Hall effect Teslameter (Lakeshore Model 475-DSP, Series 42302), equipped with the Sensor (Model HMMT-6J04-VR, Series HA2580), which has been calibrated against the reference of the Magnetic Induction Standard found in CNEA. The measurement procedure consists of measuring the transverse component (perpendicular to the axis) of the magnetic induction along the axis, from one end to the other of the working region, with a constant step that allows obtaining no less than 10 points, using a transverse hall probe. A system will be used for millimetric positioning of the hall probe with three degrees of freedom in mutually perpendicular directions (X, Y, Z axes). The results determine the average value of the magnetic induction in the work region and the standard deviation. The glass bottles with samples were placed in the center of the magnetic device to expose the cultures to the SMF, guaranteeing the same intensity of light and aeration as the control cultures. The intensity of the SMF applied to the cultures was chosen following the results obtained in previous research [1]. The SMF was applied to a 100 mL inoculum for 1 hour before being inoculated in one liter of BG110 culture medium. The control culture (C2-Control) was maintained under the same conditions (temperature, light, aeration, and nutrients). All assays were performed in triplicate.”
Changed on page 11, paragraph 4, lines 439-444“Cyanobacterial extracts were prepared at a concentration of 10 mgmL-1, 300 mg of freeze-dried biomass in 30 mL of distilled water (dH2O) and kept for 12 hours under constant stirring at 300 rpm. It was then heated at 50°C for 1 hour with the same stirring intensity [2]. The mixture was allowed to cool down to room temperature and then sonicated for 5 min. Afterward, it was centrifuged at 4500 rpm for 5 min, and the supernatant was filtered with a sterile filter. In the end, three cyanobacterial extracts were obtained.”
Comments 2: Could the authors provide more detailed information on the specific parameters used during the SMF treatment, such as duration and exact field strength.
Response 2: We agree and have accordingly changed the text. Changed on page 11 paragraph
3, lines 429-436. “The temperature conditions in the culture lab were stable at 22 ± 2.3 ℃. A LED light panel was used to keep the cultures illuminated under continuous light. The light intensity was adjusted daily with a Traceable Light Meter, maintaining the cultures with an illumination of 48.83 μE m-2s-1. Cultures were aerated by bubbling filtered air through 0.20 µm glass microfiber syringe filters. The device to aerate the cultures was an air pump for aquariums and ponds (SERA air 550 R plus, Germany) with an airflow capacity of 9.2 Lmin-1.
The SMF was applied in a 100 mL inoculum for one hour before being inoculated in one liter of BG110 culture medium.”
Comments 3: It would be beneficial for the authors to discuss how environmental factors might influence the effectiveness of biopriming with cyanobacteria and SMF.
Response 3: We acknowledge the reviewer’s concern regarding plant performance in outdoor conditions. However, our approach ensures that seedlings are first cultivated under controlled conditions, allowing them to establish and gain strength before being transplanted into open-air environments. This initial phase of controlled growth enhances their resilience and adaptability, improving their chances of successful establishment once exposed to external conditions. However, it is important to follow up further on plant growth and productivity in the field. We also added this to the end of the discussion where future studies are addressed (lines 375-385; see also response 7).
Comments 4: For the reader's interest, the presentation of data, particularly in figures and tables, should be improved for clarity. Please ensure that all figures are clearly labeled and that legends provide sufficient context.
Response 4: We have done this and hope to clearly addressed this comment. Changed: page 4, paragraph 1, table 1, line 146-147; page 4, paragraph 3, table 2, line 163-165; page 5, paragraph 4, figure 2, line 199; page 6, paragraph 2, figure 3 line 217-221; page 7, paragraph 1, table 3 line 235-236.
Comments 5: To strengthen the argument for the significance of findings and provide a broader context for their implications in agricultural practices. The discussion section could be expanded to include a more comprehensive analysis of how the observed effects of biopriming and SMF treatment relate to existing literature.
Response 5: Changed on page 7, paragraph 3, lines 250-260. “Biopriming stands out as an eco-friendly technique for increasing the rate of seed germination and overall fitness of plants by playing an important role in various events in the plant's life cycle [3]. Improvement in growth due to seed priming could be due to the release of plant growth regulating compounds by the cyanobacteria. Paul and Nair (2008) reported stimulation of plant growth due to plant growth regulating hormones and compounds such as amino acids, sugars and vitamins excreted by cyanobacteria [4]. The secretion of these compounds not only promoted growth but mitigated the adverse impact due to abiotic stress in plants [5]. Growth of plants used in dry land restoration improved due to seed biopriming with cyanobacteria [6].Osman et al. (2021) observed positive impact on the growth of wheat plants due to priming of the seeds with cyanobacterial extracts [7].”
Comments 6: It would be beneficial to outline specific experiments or approaches that could be taken to investigate these mechanisms.
Response 6: Thank you, we agree with this comment. Changed on page 10, paragraph 2, lines 375-379. “The extract of nitrofixing cyanobacteria consortium with the different treatments showed their biostimulant effect on Cucumis sativus seeds. However, the mechanism of action of these extracts, either through the action of phytohormones, other beneficial substances or the synergistic effect of both, is still unclear. Multi or meta-omics analyses may help to assess the molecular mechanisms of plant-cyanobacterial consortium interactions.”
Comments 7: It would be beneficial to discuss how environmental factors, such as soil type and climate conditions, might influence the effectiveness of biopriming with cyanobacteria and SMF.
Response 7: Your comment is very interesting and in addition to comment 3, we added this in the future perspectives to envisage experiments to assess this technology for sustainable agriculture (lines 375-379). “In addition, our approach ensures that seedlings are first cultivated under controlled conditions, allowing them to establish and gain strength before being transplanted into open-air environments. This initial phase of controlled growth enhances their resilience and adaptability, improving their chances of successful establishment once exposed to external conditions. However, future field scale studies to follow-up plant growth and productivity in the field are important to assess this contribution to sustainable agriculture.”
- Silveira Font, Y.; Ortega Díaz, Y.; Cuypers, A.; Alemán, E. I.; Vandamme, D., Influence of a static magnetic field on the photosynthetic apparatus, cell division, and biomass composition of a Chlorella microalgae-bacteria consortium. Journal of Applied Phycology 2023, 1-16. https://doi.org/10.1007/s10811-023-03137-2.
- Righini, H.; Francioso, O.; Di Foggia, M.; Prodi, A.; Quintana, A. M.; Roberti, R., Tomato seed biopriming with water extracts from Anabaena minutissima, Ecklonia maxima and Jania adhaerens as a new agro-ecological option against Rhizoctonia solani. Scientia Horticulturae 2021, 281, 109921. https://doi.org/10.1016/j.scienta.2021.109921.
- Srivastava, S.; Tyagi, R.; Sharma, S., Seed biopriming as a promising approach for stress tolerance and enhancement of crop productivity: A review. Journal of the Science of Food Agriculture 2024, 104 (3), 1244-1257. https://doi.org/10.1002/jsfa.13048.
- Paul, D.; Nair, S., Stress adaptations in a plant growth promoting rhizobacterium (PGPR) with increasing salinity in the coastal agricultural soils. Journal of basic microbiology 2008, 48 (5), 378-384. https://doi.org/10.1002/jobm.200700365.
- Poveda, J.; González-Andrés, F., Bacillus as a source of phytohormones for use in agriculture. Applied Microbiology and Biotechnology 2021, 1-17. https://doi.org/10.1007/s00253-021-11492-8.
- Chua, M.; Erickson, T. E.; Merritt, D. J.; Chilton, A. M.; Ooi, M. K.; Muñoz‐Rojas, M., Bio‐priming seeds with cyanobacteria: effects on native plant growth and soil properties. Restoration Ecology 2020, 28, S168-S176. https://doi.org/10.1111/rec.13040.
- Osman, M. E.-A. H.; Abo-Shady, A. M.; Gaafar, R. M.; Ismail, G. A.; El-Nagar, M. M., Assessment of cyanobacteria and tryptophan role in the alleviation of the toxic action of brominal herbicide on wheat plants. Journal of Crop Health 2022, 75, 785–799. https://doi.org/10.1007/s10343-022-00785-1.

Reviewer 3 Report
Comments and Suggestions for Authors
Comments for Authors:
1. The sample size (n=25 seeds per treatment group) appears relatively small.
2. While a control group (no biopriming) is included, the lack of a control for SMF application (i.e. cyanobacteria grown without SMF exposure but still used for biopriming) limits the ability to isolate the impact of SMF from other factors.
3. Results – this overall section needs to be more concise. For example:
Figures 1 & 2: Combine these into a single figure. Since these are very similar graphs, showing only one graph with both 7 and 15 days would improve the flow of the paper.
Figures 3 & 4: Combine these figures into one panel for a more compact presentation.
4. I got lost as to why Auxin concentration is presented in the beginning?
5. Line 179-184: The text does not refer to the figure.
6. Clearly state which comparisons are statistically significant based on the figures.
Author Response
Comments 1: The sample size (n=25 seeds per treatment group) appears relatively small.
Response 1: Thank you for pointing this out. Changed page 11, paragraph 6, line 456, 459-460. Germination was carried out with a sample of 100 seeds, using 25 seeds per plate with four replicates for each treatment.
Comments 2: While a control group (no biopriming) is included, the lack of a control for SMF application (i.e. cyanobacteria grown without SMF exposure but still used for biopriming) limits the ability to isolate the impact of SMF from other factors.
Response 2: Maybe, it was not clearly mentioned, but treatment C2-1 corresponds to the nitrofixing cyanobacteria consortium without static magnetic field treatment. This is always mentioned in the captions of figures and tables. Without biopriming (only H2O: control) or with biopriming (C2-1: consortium not exposed to SMF; C2-2: consortium exposed to SMF of 40-50 mT; C2-3: consortium exposed to SMF of 100-200 mT).
Comments 3: Results: This overall section needs to be more concise. For example:
Figures 1 & 2: Combine these into a single figure. Since these are very similar graphs, showing only one graph with both 7 and 15 days would improve the flow of the paper.
Figures 3 & 4: Combine these figures into one panel for a more compact presentation
Response 3: Thank you for pointing this out. We believe that figures 1 and 2 should not be joined because of the different growth conditions: (1) germination is performed until day 7 in plates and then root and stem measurements are performed. (2) Subsequently, they are planted in the soil, and measurements are taken again after 15 days.
Changes on page 6, paragraph 2, and line 211 in Figures 3 and 4 were made as suggested in one panel.
Comments 4: I got lost as to why Auxin concentration is presented in the beginning?
Response 4: The auxin concentrations are presented first as it is the auxin concentration of the cyanobacterial consortium biomass used before the extract was prepared for biopriming. It is mentioned that auxin, plant-growth stimulating hormone, is often produced by plant growth-promoting bacteria to promote germination and seedling growth [1].
Comments 5: Line 179-184: The text does not refer to the figure.
Response 5: Thank you for pointing this out. Changed page 4, paragraph 4 line 168. “In addition, the root and stem lengths were measured 7 days after germination (Figure 1).”
Comments 6: Clearly state which comparisons are statistically significant based on the figures.
Response 6: Thank you for pointing this out. We carefully checked throughout the results’ section.
- Fiodor, A.; Ajijah, N.; Dziewit, L.; Pranaw, K. Biopriming of seed with plant growth-promoting bacteria for improved germination and seedling growth. Frontiers in Microbiology 2023, 14, 1142966, doi:doi: 10.3389/fmicb.2023.114296

Round 2
Reviewer 3 Report
Comments and Suggestions for Authors
The authors have addressed my questions, and no further comment.